# Common Variable Immunodeficiency in Elderly Patients: A Long-Term Clinical Experience

**DOI:** 10.3390/biomedicines10030635

**Published:** 2022-03-09

**Authors:** Maria Giovanna Danieli, Cristina Mezzanotte, Jacopo Umberto Verga, Denise Menghini, Veronica Pedini, Maria Beatrice Bilò, Gianluca Moroncini

**Affiliations:** 1Department of Clinical and Molecular Sciences, Marche Polytechnic University, 60126 Ancona, Italy; m.b.bilo@univpm.it (M.B.B.); g.moroncini@univpm.it (G.M.); 2Department of Internal Medicine, Clinica Medica, Ospedali Riuniti, 60126 Ancona, Italy; 3Internal Medicine Residency Program, Marche Polytechnic University, 60126 Ancona, Italy; cristina.mezzanotte.1@gmail.com; 4Department of Life and Environmental Sciences, Marche Polytechnic University, 60131 Ancona, Italy; jacopoumberto.verga@gmail.com; 5The SFI Centre for Research Training in Genomics Data Science, National University of Ireland, H91 FYH2 Galway, Ireland; 6Section of Internal Medicine, Ospedale di Civitanova Marche, 62012 Civitanova Marche, Italy; denise.menghini90@gmail.com; 7Section of Internal Medicine, Department of Medicine, Carlo Poma Hospital, 46100 Mantova, Italy; ve.pedini@gmail.com; 8Allergy Unit, Department of Internal Medicine, Ospedali Riuniti, 60126 Ancona, Italy

**Keywords:** autoimmunity, biologics, cancer, clinical phenotypes, common variable immunodeficiency (CVID), elderly, genetics, immunoglobulin, inborn errors of immunity, infections, precision medicine

## Abstract

Background: Common variable immunodeficiency (CVID) is a complex, predominantly antibody deficiency usually diagnosed between 20–40 years. Few data about elderly patients are reported in the literature. Our aim was to evaluate the clinical phenotypes of elderly patients with CVID. Method: A retrospective analysis of adult patients with CVID was performed in our Referral Centre, focusing on the main differences between “older” patients (≥65 years at the diagnosis) and “younger” patients (<65 years). Results: The data from 65 younger and 13 older patients followed up for a median period of 8.5 years were available. At diagnosis, recurrent infections represented the only clinical manifestation in 61% and 69% of younger and older patients, respectively. The incidence of autoimmune diseases was higher in elderly patients compared with younger ones (30 vs. 18%, respectively). During the follow-up, the incidence of autoimmune disorders and enteropathy increased in the younger patients whereas neoplasia became the most prevalent complication in the elderly (38%). All patients received a replacement therapy with immunoglobulin, with good compliance. Conclusion: CVID occurrence in elderly patients is rarely described; therefore, the clinical characteristics are not completely known. In our series, neoplasia became the most prevalent complication in the elderly during the follow-up. In elderly patients, 20% SCIg was as safe as in the younger ones, with good compliance. A genetic analysis is important to confirm the diagnosis, identify specific presentations in the different ages, clarify the prognosis and guide the treatment. Future clinical research in this field may potentially help to guide their care.

## 1. Introduction

Common variable immunodeficiency (CVID) disorders represent the most frequent symptomatic group of inborn errors of immunity (IEI) of adulthood, consisting of a complex heterogeneous immune disorder characterised by hypogammaglobulinemia, a reduced specific antibody response and increased susceptibility to recurrent infections [1,2,3]. In addition, non-infectious complications such as autoimmune diseases, enteropathy, lymphoproliferation and malignancies affect about half of the patients [1,4,5].

A diagnostic delay represents a major issue in this setting as it can lead to the development of several complications. According to The European Society for Immunodeficiencies (ESID) Registry, the mean diagnostic delay is 8.8 years with a median of 4 years (range: 0–69 years) [2]. CVID can be diagnosed at any age with a peak in the onset of symptoms during the first three decades of life [2]. According to the recent analysis of the ESID Registry, the median age at diagnosis is 31 years (26 for males and 34 for females) [2].

In the elderly, the diagnosis may be made after a prolonged diagnostic delay or, less commonly, the onset of the first symptoms of CVID. Care of elderly patients with CVID can be challenging, requiring attention to the comorbidities of the patient and an evaluation of the disease-associated damage as well as the possible side effects of therapy and surveillance of the onset of neoplasms, which are more frequent in the older population [6]. CVID occurring in the elderly has been described in few case reports and further evidence of the characteristics of elderly patients with CVID and their clinical outcomes is required [6,7]. The aim of our work was to evaluate the main features of the elderly patients in our population of CVID adult patients.

## 2. Materials and Methods

### 2.1. Setting and Patients

Adult patients with CVID diagnosed and followed up at the Clinica Medica from the Ospedali Riuniti of Ancona and Marche Polytechnic University (Central Italy) were included in this retrospective analysis. Our centre is the referral centre in the Marche Region for IEI in adulthood as well as a regional referral centre for IPINet (Italian Primary Immunodeficiencies Network) and the Documenting Centre for the European Society for Immunodeficiencies [8].

The patients were diagnosed with CVID according to the revised European Society for Immunodeficiency (ESID) criteria [9] and/or the International Consensus Document (ICON) criteria for cases preceding the ESID criteria [1]. They were treated and followed up according to the current clinical practice relative to the time of their management. In addition to the clinical criteria (increased susceptibility to infections, autoimmune manifestations, granulomatous disease, unexplained polyclonal lymphoproliferation and an affected family member with an antibody deficiency), CVID was considered to be probable in a patient with a marked decrease of IgG and a marked decrease in at least one of the isotypes IgM or IgA (measured at least twice; <2 SD of the normal levels for age). Moreover, our patients fulfilled all the following criteria: 1: An onset of CVID, as defined by the presence of the first finding related to the disease, >4 years of age; 2: absent isohemagglutinins and/or a poor response to vaccines and low switched memory B cells (the latter was available from 2017 in 28 (35%) patients); 3: the exclusion of secondary causes of hypogammaglobulinemia (details of the differential diagnosis of hypogammaglobulinemia can be seen at https://esid.org/Working-Parties/Clinical-Working-Party/Resources/Diagnostic-criteria-for-PID2#Q5, accessed on 7 March 2022) [10]; and 4: no evidence of a T cell deficiency (available in 58 (74%) patients).

All patients provided informed consent for the data collection and publication (Prot. n: 2016 0561 OR 27 October 2016 and Prot. n: 2021 485 16 December 2021, Comitato Etico Regionale delle Marche). The study was conducted in accordance with the Good Clinical Practice guidelines, the International Conference on Harmonization guidelines and the Declaration of Helsinki for medical research involving human subjects.

### 2.2. Data Collection and Evaluation

Our patients were treated and followed up according to the current clinical practice at the time of their management. For each patient, we collected their demographic data and full anamnesis, comprising age at the onset of the first findings related to CVID, at the diagnosis and at the start of treatment as well as comorbidities and concomitant/associated drugs, familiarity for immunodeficiencies and autoimmune diseases and consanguinity among parents and grandparents.

Each patient underwent a clinical follow-up every 3–6 months, comprising a complete physical examination and anamnesis focused on infective recurrences and on a new appearance or the progression of disease complications.

Blood tests at the first visit included both routine ones as well as more specific exams aimed at confirming the CVID diagnosis, excluding a secondary immunodeficiency (i.e., haematological disorders) and predicting possible complications. At the first evaluation, all patients underwent the following blood tests:A complete blood count (CBC), hepatic and renal function indexes, serum electrophoresis, urine analysis, C reactive protein (CRP) and erythrocyte sedimentation rate (ESR);IgG, IgA, IgM, IgE and IgG subclasses serum levels, serum and urine immunofixation, serum and urine free light Ig chains, anti-tetanus antibodies at the baseline and after a vaccination with tetanus toxoid (when indicated);T and B cell subsets (assessed using a flow cytometer analysis) focusing on the B cell maturation pathway;serological screening for HIV, EBV, CMV, HBV, HCV and H. pylori stool antigen research;Coombs test, D-dimer, ANA, anti-phospholipid antibodies and Lupus anticoagulant (LAC).

Other exams (i.e., C3–C4, anti-ENA antibodies, malabsorption indexes, faecal calprotectin, TSH, lactate dehydrogenase and beta-2 microglobulin) were selected according to the clinical condition of the patient. The timing for the laboratory follow-up was based on the clinical conditions and immunoglobulin serum values. Generally, the patients were tested for Ig levels and a routine analysis every 3–6 months and every 12–24 months for the exclusion of emerging causes of a secondary immunodeficiency. The surveillance oncological programs were those as in age- and sex-matched populations. Due to the impact of neoplasia on the survival of patients, the early diagnosis of cancer must receive great attention; thus, we performed strict surveillance programs as previously outlined [11,12].

In selected cases, we also performed a high-resolution chest computed tomography (CT) scan, a colonoscopy with a biopsy and a thyroid ultrasound. Pulmonary function tests, abdominal ultrasounds and oncological screening tests were performed in all cases each year. In other cases, the repetition time was defined according to the clinical condition of the patient.

A profile with the B cell subpopulations (classical naïve (CD19+CD27−IgD+), switched memory (CD19+CD27+IgD−) and unswitched (CD19+CD27+IgD+)) was available only after 2017; therefore, it was not possible to distinguish all patients of our series according to the Freiburg classification [13]. Unfortunately, it was not possible to study the CXCR5+ subset of the CD4+ T cells, usually referred to as the circulating TFH (cTFH) cells.

The data collected from patients ≥ 65 years at the time of CVID diagnosis (the group of “older” patients) were compared with the data obtained from patients with a CVID diagnosis before the age of 65 years (the group of “younger” patients). The variables considered were the age at the onset of symptoms and at the diagnosis, diagnostic delay (interval between the first symptoms and the diagnosis), Ig levels at the diagnosis, replacement therapy and clinical phenotype percentage at the diagnosis and follow-up in both groups. The clinical phenotypes were defined according to Chapel et al. [14]. We recorded the data related to replacement therapy (RT) with human immunoglobulin (Ig). Patients received intravenous immunoglobulin (IVIg from a regional donor blood bank), 20% subcutaneous immunoglobulin (20% SCIg, Hizentra^®^ CSL Behring, King of Prussia, PA, USA) or facilitated 10% SCIg (fSCIg, Hyqvia^®^ Takeda, Japan) at a monthly dose of 0.4–0.6 g/kg. The dose adjustments were based on infective recurrences and the serum IgG levels (target serum IgG levels were around 700–800 mg/dL). For the subcutaneous Ig, we trained patients and a caregiver to be able to perform a home treatment. The first three infusions were performed in a hospital under medical supervision; when the patient felt confident to conduct self-administration and how to recognise and face adverse events, they could continue at home.

### 2.3. Statistical Analysis

All variables of interest were calculated through descriptive statistics. The categorical data were presented as frequencies and percentage values whereas the continuous variables were presented as median values and their relative range. The Mann–Whitney non-parametric test was used to compare independent groups, depending on the nature of the distribution of the data under study. A *p* < 0.05 was considered to be statistically significant. All analyses were carried out with SPSS (SPSS version 21.0, IBM, Armonk, NY, USA).

## 3. Results

### 3.1. Baseline Characteristics

Data from 78 patients were available for the analysis: 65 younger (<65 years old) and 13 older subjects (≥65 years old) followed up for a median period of 8.5 years (range: 0–144 months). Younger patients were diagnosed and followed up from 2008 whereas we enrolled the first older patient in 2014.

Table 1 shows the main characteristics of the CVID patients at the diagnosis. The female gender prevailed in both populations. The median diagnostic delay was 84 and 36 months in younger and older patients, respectively, whereas the median therapeutic delay was 96 and 38 months in younger and older patients, respectively. The serum Ig levels are depicted in Table 2. We did not document a protective response to tetanus toxoid in 62 (79%) patients.

Among the older patients, only one (female) reported a consistent family history (a son with CVID and a nephew with autoimmune thyroiditis). In the younger group, 13 (20%) patients referred to a family history (autoimmune diseases (AID) in 5 cases, a predominant antibody deficiency (CVID, hypogammaglobulinemia of uncertain significance and selective IgA deficiency) in 4, both AID and PAD in 2 and both cancer and AID in 2).

### 3.2. Clinical Features at Disease Onset

In the younger group, all patients had recurrent upper airway infections at the diagnosis, which represented the only clinical manifestation in 40 (61%) patients of this group (Table 3). In the remaining 25 patients, we detected 13 autoimmune complications in 12 (18%) including idiopathic thrombocytopenic purpura (ITP) in 7 and autoimmune haemolytic anaemia (AIHA) in 2 as well as insulin-dependent diabetes mellitus (IDDM), systemic sclerosis, neuromyelitis optica (Devic’s disease) and psoriatic arthritis. Three patients had lymph node granulomatous disease (5%), six patients presented enteropathy (9%) and neoplasia was detected in four (6%).

In the older group, nine patients (69%) had only recurrent upper airway infections at the diagnosis (Table 3). Four patients (31%) reported an autoimmune complication at the disease presentation (AHIA, recurrent myelitis, psoriatic arthritis, and vasculitis). We documented polyclonal lymphoproliferation, enteropathy and neoplasia in one case each.

### 3.3. Clinical Features at Last Follow-Up Control

Table 3 shows the manifestations recorded during the follow-up in the two groups. In the younger group, we documented a new complication as an autoimmune disease in 17 patients, polyclonal lymphoproliferation in 10, enteropathy in 6 and neoplasia in 9. At the last follow-up, 18 patients had overlapping features.

During the follow-up, four older patients presented new onset complications: polyclonal lymphoproliferation (*n* = 1) and ITP (*n* = 1); in both cases, these were associated with neoplasia (bladder cancer and lymphoma, respectively). Two more older patients received a diagnosis of cancer (gastric schwannoma and non-melanoma skin cancer).

A splenectomy was performed on six younger patients (9%); conversely, no elderly subjects underwent this procedure.

In addition to CVID non-infectious complications, other main CVID-uncorrelated comorbidities were considered such as systemic arterial hypertension, diabetes mellitus and dyslipidaemia. We did not detect any relevant differences between the young and elderly groups.

### 3.4. Occurrence of Neoplasms

Overall, 13 younger patients (20%) and 5 older patients (38%) reported neoplasia. In the younger group, cancer represented the first clinical manifestation in four patients whereas in the older group, this was only in one patient. During the follow-up, nine younger patients and four older patients developed neoplasia.

In the younger patients, we observed lymphoproliferative disorders (*n* = 6), gastric cancer (*n* = 2), breast cancer (*n* = 2), thyroid cancer (*n* = 2), pancreatic cancer (*n* = 1) and melanoma (*n* = 1). One patient presented two different neoplasms. Three of them died from the tumour progression.

In the older patients, we observed lymphoproliferative disorders (*n* = 2), gastric schwannoma (*n* = 1), non-melanoma skin cancers (*n* = 1) and bladder cancer (*n* = 1); one elderly patient died from the tumour progression.

The malignancies were treated according to the specific oncological protocol and guidelines in force at that time.

### 3.5. Serum Ig Levels and Immunological Phenotype

For the laboratory parameters, we documented lower serum IgG levels at the diagnosis in the older patients (median IgG levels: 333 mg/dL in the younger vs. 270 mg/dL in the elderly) (Table 2). We did not find any relevant difference in the serum IgA or IgM levels in the patients with CVID younger than 65 years compared with the older ones. Regarding the immunological phenotype, compared with the control subjects, there were no differences in the absolute count and percentage of lymphocytes, T cell subsets and CD19+ B cells (data available from 58 (74%) patients). We detected a peripheral B cell count < 1% of the total lymphocytes in one patient in each group. Immunophenotyping with a study of the B cell maturation was available from 2017 in 28 (35%) patients and demonstrated low switched memory B cells (CD19+CD27+IgD−) in 75% of them. The patients who underwent immunophenotyping showed comparable results with the Freiburg analysis; about 75% of them were included in group I (CD27+IgM−IgD− < 0.4%) without a further distinction in Ia or Ib (the CD21 low B cell expression was not tested) as well as the remaining 25% of the patients in group II (CD27+IgM−IgD− > 0.4%).

### 3.6. Treatment

All patients were treated with Ig RT with three different modalities according to the clinical conditions and preferences of the patients. In the younger group, 20 (30%) patients were treated with IVIg, 25 (38%) with 20% SCIg and 18 (27%) with fSCIg. In this group, due to a previous severe reaction to IVIg, two patients received an antibiotic prophylaxis (oral azithromycin, 250 mg/d for three days/weekly). In the older group, all patients received IgRT: IVIg in 7 (54%) patients and 20% SCIg in 6 (46%).

All patients reached protective serum IgG levels with a marked reduction of recurrent infections. For the subcutaneous route, the elderly patients self-administered the prescribed infusions at planned doses at home. No premedication was used. The adherence to the treatment was 100%. We did not detect any major side effects related to the 20% SCIg treatment. Only local adverse events were reported; these were mild and self-limiting such as redness and/or irritation at the site of infusion, which disappeared over time.

The patients underwent specific treatments other than Ig replacement therapy to control the CVID complications. Patients affected by autoimmune cytopenia received high doses of glucocorticoids (prednisone 1 mg/kg/d or intravenous boluses of 500 mg or 1 g i.v. in severe cases). In the past, patients underwent a splenectomy in refractory cases. Other autoimmune complications were treated similarly to non-CVID-related rheumatological diseases with low–medium doses of glucocorticoids and disease-modifying antirheumatic drugs (DMARDs), i.e., methotrexate, azathioprine and hydroxychloroquine, with a good clinical outcome, whereas adalimumab is an anti-TNF-α agent.

### 3.7. Prognosis

Over the median 8.5-year follow-up period, five younger (8%) and two older patients (15%) died. The causes of death in the first group were neoplasia (*n* = 3), sepsis (*n* = 1) and an intracerebral haemorrhage (not correlated to CVID, *n* = 1); in the other group, septic shock, and complications due to neoplastic disease (bladder cancer) were responsible.

The median age at death was 57 and 81 years for young and older patients, respectively; in both cases, the age at death was lower than that of the gender-matched population in our Marche Region.

## 4. Discussion

We present here our data related to CVID in patients aged ≥ 65 years, representing 16% of our series, with the aim of expanding existing evidence of CVID in an elderly population.

There are few data reported in international registers [2,15,16]. The ESID reported 9.7% of diagnoses in subjects >60 years old [2]; a similar percentage emerged from the IPINet registry in 2007 [15]. In both registries, the onset of symptoms was typically earlier with a median diagnostic delay of about 8–9 years. In the ESID registry, less than 50% of newly diagnosed elderly patients (>60 years old) had an onset of symptoms at the same time as the diagnosis [2]. Therefore, a diagnostic delay can explain at least part of the CVID occurrence in elderly patients, even if a substantial number of subjects develops first symptoms when >60 years old. A diagnostic delay is associated with the development of several CVID-related complications such as chronic lung disease and other malignancies. A diagnostic delay is strongly associated with a therapeutic delay (related to the beginning of Ig replacement therapy). An earlier diagnosis of CVID allows earlier IgG replacement therapy, which reduces the susceptibility to infections, preventing or delaying end-organ changes linked to infections such as bronchiectasis. Regarding malignancies, in a previous work of our group, we found that patients who developed a malignancy had a longer diagnostic delay (and therapeutic delay) in comparison with patients with no malignancy [17].

In our series, the older patients had a shorter diagnostic delay compared with the younger ones. This could be explained by a closer clinical follow-up or a prompt investigation in the case of the detection of hypogammaglobulinemia in the elderly. We diagnosed CVID in elderly patients in more recent years (from 2014) in comparison with the younger ones; therefore, it is possible that this could be due to an increased awareness of the immunodeficiencies and/or to a faster referral to a specialised centre.

Regarding the symptoms at the disease onset, the IPINet Registry described a significant increase of chronic lung disease and chronic diarrhoea in the elderly group (>50 years old) in comparison with the younger patients [15]. Otherwise, splenomegaly and autoimmune phenomena appeared to be independent of age [18]. In our series, patients presenting with one or more clinical manifestations in addition to recurrent infections (i.e., “complicated” patients) at the onset were almost equally represented in both groups (39% older vs. 31% younger, respectively). However, we detected a few differences in the type of presentation between the two groups. At the diagnosis, the incidence of autoimmune diseases was two-fold in the elderly patients compared with the younger (30 vs. 18%, respectively). During the follow-up, the incidence of autoimmune diseases (from 18 to 44%, respectively) and enteropathy (from 9 to 18%, respectively) increased steadily in the younger patients whereas in the elderly, neoplasia became the most prevalent complication, concerning 38% of patients. In recent years, complications related to autoimmune conditions and neoplasia have increasingly been reported in CVID patients [3,14,19,20]. Specific treatments for non-infectious complications of CVID include high doses of glucocorticoids and/or a splenectomy (in the past) for autoimmune cytopenia. Methotrexate, azathioprine, adalimumab, and hydroxychloroquine have been employed according to the main recommendations [21] with benefits. It is important to note that a high-dose IVIg or a facilitated subcutaneous Ig (fSCIg) have been used as immunomodulatory agents and not only as a replacement therapy.

Globally, neoplasia affected 23% of our patients (18/78). These data are similar to those reported in a multicentre Italian study [12]. In our series, the mean age at the diagnosis of neoplasia was 54 years, which was in agreement with the same study; this was 59 years old for females and 41 years old for males. These findings differed from those described in the 2019 Report on the Health of Marche Region [22] in which the mean age at neoplasia diagnosis was higher; 67 years old in females and 69 years old in males. In CVID, the most frequent neoplasia observed, as widely reported in literature, was lymphoma and gastric and breast cancer among the solid tumours; in our series, thyroid cancer also exhibited the same prevalence as gastric and breast cancer [12,23]. In contrast, in the Marche Region, the most frequent tumours observed were colon, breast, lung and prostate cancer.

Seven patients in the global cohort died (7/78 = 9%). Neoplasia represented the leading cause of death in our population, in agreement with the most recent findings and in the younger group [5,12]. Neoplasms causing death in the younger population were non-Hodgkin lymphoma, gastric and pancreatic cancer whereas one elderly patient died of bladder cancer. In the Marche population, neoplasia represented the second cause of death after cardiovascular diseases, accounting for 27% of the global mortality. In 2016, the neoplasms causing the most deaths in the Marche region were lung, colon, pancreas, stomach, and breast cancer, in descending order.

At the diagnosis, a phenotyping of the peripheral blood lymphocytes was performed in 74% of patients and there were no differences between the younger and elderly groups or between the different clinical phenotypes. Except for two cases, the absolute count of B lymphocytes was in the normal range. The 28 (35%) patients who underwent immunophenotyping in 2017 or later showed a delay in their maturation and consequent accumulation of “naive” and “unswitched” as well as a lack of “switched memory” cells. We did not find significant differences between the young and elderly patients based on the Freiburg classification [13]. Moreover, we did not find any significant correlation between the I and II Freiburg groups and the development of CVID-related complications. In the patients with CVID-associated lymphoproliferative diseases, there were alterations in the absolute count and phenotypic profile of the subpopulations of T and NK lymphocytes.

For the treatment, the patients received Ig replacement therapy and prophylactic azithromycin [24]. Although IVIg has been shown to be usually safe in older patients [25], IVIg administration can be associated with adverse events (i.e., acute renal failure and thrombotic complications) with a concern for old and pluri-comorbid patients. In comparison, a 20% SCIg administration has a better safety profile in elderly subjects independent of comorbidities and associated drugs (i.e., anticoagulants and antiplatelet agents) [26]. Approximately half of our elderly patients received 20% SCIg at home with no administration problems, even at the long-term follow-up [27]. None of these patients experienced side effects leading to a discontinuation/modification of the treatment schedule. Despite the reported skin fragility of elderly people, we did not document this issue. Verma et al. [6] reported a concern about possible technical difficulties for self-infusion at home, which we did not document. An adequate knowledge of the family context and the identification of a caregiver are essential for the selection of candidates for subcutaneous treatment among elderly subjects. Therefore, our experience confirmed that a 20% SCIg treatment could be safely performed in elderly patients with a good adherence to the prescribed treatment.

Nowadays, a molecular analysis has become significantly relevant for better understanding and treating patients affected by inborn errors of immunity (IEI) [28]. Thanks to the introduction of next generation sequencing (NGS) methods in the last ten years, more than 400 gene defects implied in IEIs have been identified [29]. Regarding CVID, only 20% of cases are caused by monogenic defects; the remaining 80% derives from polygenic variants and/or from epigenetic alterations [30,31]. Interestingly, a study recently published by Khandia et al. considering a panel of 42 genes involved both in PID and cancer highlighted how these genes are under a selection pressure [32]. These cases can be successfully identified crossing whole genome sequencing (WGS) or whole exome sequencing (WES) with RNAseq methods to discover gene–gene interactions and expression dysregulations not detectable by genomic studies only [33,34,35]. In the workup of patients with CVID, analyses by WES and WGS are not routinely performed at our centre. Consequently, we only performed a genetic diagnosis for a few selected patients with a particular phenotype or family history (n = 12). Among them, two patients had TACI mutations and another patient had a CTLA-4 mutation. This investigation was then interrupted due to the COVID-19 pandemic. Analysing our data and the related literature, we could speculate about the two cohorts of CVID patients: the first one composed of subjects with an early onset; and the second one including those with a late onset of hypogammaglobulinemia and related diagnosis [17,35,36]. This second cohort could reasonably include patients with no genetic defects in the coding DNA regions, but with alterations regarding the expression regulation and chromatin organisation coming from mutations in the regulatory regions and epigenetic modifications. An increasing number of studies has used new ‘omics’ approaches such as methylome, histone modification mapping and miRNome to investigate the differences in gene regulation among CVID patients and healthy controls [33]. In a cohort of late-onset CVID patients, Jorgensen et al. found increased methylation levels in B and T cells, suggesting an epigenetic role in lymphocyte maturation [37].

There is also an increasing interest in non-coding RNA (ncRNA), which can alter the gene expression and chromatin structure [38]. Rae et al. focused attention on micro-RNAs (miRNAs such as miR-142 and miR-155) involved in lymphocyte regulation, the disruption of which can be responsible for the appearance of CVID-like phenotypes [39]. For miR-155 and miR-29, other authors have documented age-related functions, suggesting a link between late-developing CVID and epigenetic alterations [40].

A few cases of monogenic CVID can be successfully treated with an individualised target therapy. In a younger group of patients with a complex phenotype including CVID, thrombocytopenia, psoriatic arthritis and enteropathy, the detection of a mutation in CTLA4 prompted us to treat them with abatacept, with benefits [41]. Further studies are necessary to clarify the aetiology of various CVID-associated phenotypes and confirm the epigenetic impact on immunodeficiencies. Detecting the environmental factors involved in epigenetic alterations could be useful to elaborate more specific diagnostic markers and more effective therapies, implementing personalised medicine in CVID.

## 5. Conclusions

In conclusion, CVID must be suspected even in older patients because the onset could occur at any age. A diagnostic delay can explain only part of the CVID occurrence in elderly patients because a significant number of subjects developed their first symptoms when >65 years old. When analysing the clinical features of the elderly group in contrast with the younger one, at the onset of the disease almost 1/3 of the elderly patients had an autoimmune disease. Neoplasia became the most prevalent complication during the follow-up and represented the leading cause of death in our cohort. Ig RT is as safe in elderly patients as in the younger ones and should be recommended in older patients to prevent recurrent infections. A subcutaneous route of administration could be safely employed in these patients, with a good compliance. Our study could be improved by larger studies with larger cohorts as available, for example, through IPINet in Italy.

## Figures and Tables

**Table 1 biomedicines-10-00635-t001:** Baseline characteristics in 78 younger and older CVID patients.

		<65 Years Old *n* (%)	≥65 Years Old *n* (%)
Number of Patients	Male	24 (37)	2 (15)
	Female	41 (63)	11 (85)
Age at First Clinical Presentation (Years)	Mean ± SD ^1^	29 ± 18	67 ± 5
	Median	30	66
Age at Diagnosis (years)	Mean ± SD ^1^	41 ± 15	70 ± 5
	Median	41	69
Diagnostic Delay (Months)	Mean ± SD ^1^	139 ± 173	34 ± 41
	Median	84	36

^1^ SD: standard deviation.

**Table 2 biomedicines-10-00635-t002:** Serum immunoglobulin levels (mg/dl and g/L) at diagnosis in 78 younger and older CVID patients.

		<65 Years Old *n* (%)	≥65 Years Old *n* (%)
Serum IgG Levels	Mean ± SD ^1^ (mg/dL)	333 ± 135	270 ± 129
	Median g/L (IQR ^2^, g/L)	3.7 (2.6–4.4)	2.7 (1.8–3.5)
Serum IgA Levels	Mean ± SD ^1^ (mg/dL)	41 ± 44	59 ± 53
	Median g/L (IQR ^2^, g/L)	0.2 (0.08–0.5)	0.4 (0.08–0.9)
Serum IgM Levels	Mean ± SD ^1^ (mg/dL)	42 ± 41	30 ± 30
	Median g/L (IQR ^2^, g/L)	0.2 (0.1–0.6)	0.2 (0.08–0.3)

^1^ SD: standard deviation; ^2^ IQR: interquartile range.

**Table 3 biomedicines-10-00635-t003:** Clinical phenotypes at CVID diagnosis and at last follow-up control.

	<65 Years Old *n* (%)		≥65 Years Old *n* (%)	
	Diagnosis	Follow-Up	Diagnosis	Follow-Up
Non-complicated (infections only)	40 (61)	21 (32)	9 (69)	5 (38)
Autoimmunity	12 (18)	29 (44)	4 (31)	5 (38)
Immune thrombocytopenic purpura	7 (10)	10 (15)	0 (0)	1 (7)
Autoimmune haemolytic anaemia	2 (3)	2 (3)	1 (7)	1 (7)
Others (autoimmune hepatitis, Devic’s disease, Hashimoto’s thyroiditis, IDDM, myelitis, PBC, psoriasis, psoriatic arthritis, Sjogren’s syndrome, systemic sclerosis, vasculitis, vitiligo)	4 (6)	17 (26)	3 (23)	3 (23)
Polyclonal lymphoproliferation	3 (5)	13 (20)	1 (7)	2 (15)
Enteropathy	6 (9)	12 (18)	1 (7)	1 (7)
Neoplasia	4 (6)	13 (20)	1 (7)	5 (38)
LNH	2 (3)	4 (6)	1 (7)	2 (15)
Other neoplasms (stomach, pancreas, breast, skin, thyroid, LGL, bladder)	2 (3)	9 (13)	0 (0)	3 (23)

## Data Availability

Not applicable.

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
