# Peer review of "Common Variable Immunodeficiency in Elderly Patients: A Long-Term Clinical Experience"

_biomedicines, 2022, doi:10.3390/biomedicines10030635_

Round 1

Reviewer 1 Report

M.G. Danieli and co-authors report in their manuscript differences between CVID patients ≥ or < 64 years of age.

The < 65 cohort consisted of  24 males and 41 females with an mean age of 29 for first clinical presentation and 41 for the age of diagnosis. In the ≥65 year cohort there were 2 males and 11 females. The clinical symptoms started in average at 67 years of age while the diagnosis of CVID was made at 70 years of age. Serum immunoglobulin titers were similar. Clinical manifestations were comparable, except for neoplasia, which seemed to be more frequent in the elderly cohort during the observation period.

Since the cohort size is very small, it is difficult to draw any more general conclusions from this study. It therefore may serve as a "teaser" for larger studies with larger cohorts as e. g., available through IPINet in Italy.

The paper would profit from showing the composition of B and T cell subsets including CD21low B cells and circulating Tfh cells The discussion can be shortened by omitting the NGS part (lines 267 - 304)

Reviewer 2 Report

This manuscript by Danieli et al., provides new insights in to the incidence of common variable immunodeficiency (CVID) in the elderly, a population where the implications of this diseases are under recognised. This provides important information about the immunophenotype and Ig replacement therapy status of patients. However, minor edits are required.

General:

  1. Check spelling and grammar
  2. Change “primary antibody deficiency” to “predominantly antibody deficiency” in line with IUIS classification scheme.
  3. Replace commas for dots for all measurements e.g., change “0,4-0,6 g/kg” to “0.4-0.6 g/kg”.

Introduction:

  1. Include Slade et al., 2018 Front Immunol (PMID:29867917) and Resnick et al., 2012 Blood (PMID: 22180439) a reference for lines 43-44.
  2. The authors mention that diagnostic delay results in development of several complications. Give examples. What are the outcomes of these complications if untreated? Death?

Materials and Methods:

  1. Authors write that data regarding “familiarity for immunodeficiencies” was collected from each patient. What is meant by this? Do you mean family history, please correct to this and this is more familiar terminology?
  2. Were patients questioned to ensure that possible secondary causes of immunodeficiency were ruled out? If so ensure that this is mentioned in the data collection and evaluation section. Ruling out secondary causes are essential in this age group.
  3. What blood tests were performed? (Lines 83-84).
  4. What is meant by phenotype percentage? Please clarify. (Lines 99-100)
  5. Lines 101-103: The product prescribed to patients on SCIg are detailed, please specify IVIg brands.

Results:

  1. It is known that most CVID patients have coexistent non-infectious complications. This should be addressed for this cohort.
  2. Combine tables 3 and 4 so that readers can easily compare the clinical phenotypes between diagnosis and last follow up.
  3. What is the median time between diagnosis and last follow-up this should be included as it has a bearing on the emergence of symptoms.
  4. Authors mention in the methods that blood phenotyping was undertaken. What were the outcomes of this analysis? Were there differences for example in B or T cell numbers, memory B cell numbers?
  5. Were patients on treatments other than IgRT? This is important to address as the inadequacies of treatment are what leads to further complications in those with non-infectious diseases.

Discussion:

  1. Add reference PMID:32801365 to line 271-272 ending “and/or epigenetic alterations”.
  2. What is meant by the peculiar codon composition under selection pressure? Reward. (Line 274)
  3. More references should be added throughout to support statements within the manuscript especially in the discussion where multiomics is described.
  4. Line 269-270 mentions that 400 gene defects are implied in IEIs. Update to reflect the statistics in the most recent iteration of the IUIS IEI classification authored by Tangye et al., 2019 (PMID: 31953710).

Reviewer 3 Report

This is an important topic but the study has a number of drawbacks.

Major:

ESID criteria require immunological data for CVID: poor antibody response or low switched memory B cells. This is not presented. How do we know the diagnosis?

Were any genomic studies performed? CVID-like disorders are frequently identified as monogenic diseases.

Further immunophenotyping is increasingly helpful in CVID diagnosis and prognostication. There are specific subtypes. It would be great to know how it differs in the elderly, i.e. comparison of the immunophenotypes, (e.g. as in Freiburg classification). This is a big gap in the paper.

Minor:

No data on family history.

No explanation of oncological screening. I would like to know CLL, MM or MGUS were adequately excluded.

More data on comorbidities would be usueful.

Round 2

Reviewer 3 Report

There are 78 CVID patients in the study.

I still need to know if immunological criteria were fullfilled for all of them.

ESID criteria require immunological data for CVID: poor antibody response or low switched memory B cells. This should be made obvious and said /presented that every patient has such abnormality.

How many patients did have detaled immunophenotyping? 

Summary for that should be available at least in a supplement. 

Author Response

We thank Reviewer 3 for his useful observations. We corrected the text in the Materials and Methods and in Results Section. We humbly believe that our manuscript now could be suitable for publication into Biomedicines.

Patients were diagnosed with CVID according to the revised European Society for Immunodeficiency (ESID) criteria (available at: https://esid.org/content/download/17141/463543/file/ESID%20Clin%20Crit_omim_orpha_hpo_11_2019fin.pdf), and/or to the International Consensus Document (ICON) criteria, for cases preceding ESID criteria [1]. They were treated and followed-up according to the current clinical practice relative to the time of their management. Besides the clinical criteria (increased susceptibility to infection, autoimmune manifestations, granulomatous disease, unexplained polyclonal lymphoproliferation, and affected family member with antibody deficiency) CVID was considered probable in a patient having a marked decrease of IgG and a marked decrease in at least one of the isotypes IgM or IgA (measured at least twice; <2SD of the normal levels for age). Moreover, our patients fulfilled all the following: 1. Onset of the CVID, as defined by the presence of the first finding related to the disease, was >4 years of age in all patients. 2. All had ab-sent isohemagglutinins and/or poor response to vaccines and low switched memory B cells (this latter available from 2017 in 28 (35%) of patients). 3. Secondary causes of hypogammaglobulinemia have been excluded (see details in Differential diagnosis of hypogammagammoglobulinemia in https://esid.org/Working-Parties/Clinical-Working-Party/Resources/Diagnostic-criteria-for-PID2#Q5). 4.  No evidence of T-cell deficiency (available in 58 (74%) of patients).

Regarding the immunological phenotype, compared with control subjects, there were no differences in lymphocytes absolute count and percentage, T cells subsets and CD19+ B cell (data available in 58 (74%) patients). We detected peripheral B cells count <1% of total lymphocytes in one patient in each group. Immunophenotyping with study of the B cells maturation, available from 2017 in 28 (35%) patients, demonstrated low switched memory B cells (CD19+CD27+ IgD−) in 75% of them. Patients who underwent immunophenotyping showed comparable results with Freiburg analysis: about 75% of them was included in group I (CD27+ IgM- IgD- <0.4%) without further distinction in Ia or Ib (CD21low B cells expression was not tested) and remaining 25% of patients in group II (CD27+ IgM- IgD- > 0.4%).
